# A Comparative Study of Design of Active Fault-Tolerant Control System for Air–Fuel Ratio Control of Internal Combustion Engine Using Particle Swarm Optimization, Genetic Algorithm, and Nonlinear Regression-Based Observer Model

**Turki Alsuwian [1]**, **Muhammad Sajid Iqbal [2]**, **Arslan Ahmed Amin [2],***, **Muhammad Bilal Qadir [3]**, **Saleh Almasabi [1]** and **Mohammed Jalalah [1,4]**

[1]  Department of Electrical Engineering, College of Engineering, Najran University, Najran 11001, Saudi Arabia
[2]  Department of Electrical Engineering, FAST National University of Computer and Emerging Sciences, Chiniot Faisalabad Campus, Chiniot 35400, Pakistan
[3]  School of Engineering & Technology, National Textile University, Faisalabad 37610, Pakistan
[4]  Promising Centre for Sensors and Electronic Devices (PCSED), Advanced Materials and Nano-Research Centre, Najran University, P.O. Box 1988, Najran 11001, Saudi Arabia
*   Correspondence: arslan.amin@nu.edu.pk

**Abstract:** In this article, three distinct strategies for designing an Active Fault-Tolerant Control System (AFTCS) for Air-Fuel Ratio (AFR) control of an Internal Combustion (IC) engine in a process plant to avoid engine shutdown, are presented. The proposed AFTCS employs a Genetic Algorithm (GA), Particle Swarm Optimization (PSO), and a Nonlinear Regression (NLR)-based observer model in the Fault Detection and Isolation (FDI) unit for analytical redundancy. A comparison between these three proposed techniques is carried out to determine the least expensive and most accurate approach. The results show that the nonlinear regression produces highly accurate results by consuming very low computational power, and its response time is also very low as compared to GA and PSO. The results obtained show that NLR requires 99.6% and 93.1% less computational time for throttle and MAP estimation, respectively, by reducing the estimation error to as low as 0.01. The simulation of the proposed system is carried out in the MATLAB/Simulink environment. The results prove the superior fault tolerance performance for sensor faults of the AFR control system, especially for the Manifold Absolute Pressure (MAP) sensor in terms of less oscillatory response as compared to that reported in existing literature.

**Keywords:** active fault-tolerant control; AFR control; fault detection and isolation unit; IC engine; nonlinear regression; particle swarm optimization; genetic algorithm; observer model

## 1. Introduction

A fault is a discrepancy between the actual and expected value of a parameter. Fault tolerance refers to a system's capacity to continue operating under malfunctioning conditions [1]. Any real-world system can develop faults, lowering the system's dependability and performance [2]. Fault-Tolerant Control (FTC) is a technique that may be used to improve the dependability of important systems such as nuclear power plants and airplanes [3]. In faulty conditions, the system's performance can be lowered, although failure can be tolerated to some amount, provided stability is ensured. Fault-tolerant control systems are defined as systems with a high level of dependability [2]. FTC approaches are currently being applied in important industrial processes, such as combustible fuel and gas, petrochemicals, and fertilizers, where output losses cannot be allowed, and constant system efficiency is essential [2]. The active and the passive form are the two main types

of FTC [4]. The Active Fault-Tolerant Control System (AFTCS) is responsible for recognizing and isolating the system's defective components. Fault Detection and Isolation (FDI) employs the observer theory, in which a plant parameter is compared to a predetermined normal value to produce a residual [5]. If the residual is below the permissible limits, it means the machine is free of defects. If the residual exceeds the required threshold, the FDI unit reports faulty conditions. The controller is then reprogrammed to meet the current operational requirements.

Internal Combustion (IC) engines are common in the process sector and maintaining adequate Air–Fuel Ratio (AFR) control in their fuel system is vital for improved engine performance, fuel efficiency, and environmental safety. Because faults in the sensors of the AFR system cause the engine to shut down, fault tolerance is required. The fault tolerance in IC engines can be introduced with the help of analytical redundancy. The main principle of analytical redundancy is to compare real system behavior to the model [6]. The estimated sensor values can be computed by using several optimization and machine learning techniques. In this paper, we have used three techniques, namely, Genetic Algorithm (GA), Particle Swarm Optimization (PSO), and a Nonlinear Regression (NLR) model, to compute the estimated sensor values in the case of a fault. The advantages of using these techniques and the results obtained are discussed in the upcoming sections.

The rest of the paper is organized as follows: the literature review is presented in Section 2, research methodology is presented in Section 3, estimation techniques used are discussed in Section 4, and the results are discussed in Section 5, followed by a conclusion at the end.

## 2. Literature Review

Kalman filters, fuzzy logic, neural networks, regression, and other techniques may be used to construct an FDI unit. In the case of sensor and actuator faults coexisting, Kalman filters are used in [7] for Fault Detection and Location (FDL) in a nonlinear model of an aeroengine. The proposed system uses hybrid Kalman filters and only detects the faults and does not estimate the sensor values in case of a fault. Another Kalman filter-based approach is proposed in [8], where a hybrid FTCS is used to introduce analytical redundancy in an engine to maintain the Air–Fuel Ratio (AFR). An active FTC system is proposed in [9], in which an Adaptive Interval Observer (AIO)-based FDI is used to accomplish trajectory tracking of an unmanned underwater vehicle that is susceptible to sensor failures and many uncertainties. The proposed AIO is inefficient in terms of computation power. In [10], the FDI uses fuzzy logic to predict nonlinear functions, and adaptive control is used to correct bias and obtain faults in the actuator. The proposed system may accept inaccurate data and inputs; hence, the accuracy may be compromised. Ding and Fang [11] suggested an FDI method that manages sporadic observations if the full measurements of data are unavailable. A linear regression-based robust AFTCS approach can be found in [12] for the AFR control of an IC engine. In the proposed approach, the nonlinear sensor values are estimated by using linear regression; hence, the system produces less accurate results. Yu et al. [13] investigated the output degradations caused by incipient faults and predicted the residual life in a mechatronic device in a case study. Two analytical redundancy connections were discussed; these would boost the controlled system's separation capacity when multiple faults occurred. Reference [14] proposes an adaptive fault-tolerant approach for synthesizing a compensation controller for uncertain nonlinear pure-feedback systems with dead-zone actuators and stochastic failures. The failure mode of the actuator is characterized by a scalar Markovian-type function, and the compensation strategy is designed using adaptive backstepping methods.

In [15], an Artificial Neural Network (ANN)-based approach is used to estimate the sensor values in case of a fault to prevent the engine shutdown, but the critical points (on which the estimation error may be high) are excluded from the analysis. The topic of command filtering-based event-triggered adaptive fuzzy control for a class of stochastic nonlinear systems with stochastic failures and input saturation is investigated in [16]. Fuzzy

Logic Systems (FLSs) are used to approximate unknown nonlinear functions and system dynamic changes induced by stochastic errors. The command filtering design approach is used to decrease the computational overhead. The average dwell-time technique is used to examine the AFTCS problem for a class of switched nonlinear systems in [17], in which neural networks are used to design the controller. Reference [18] describes a fault-tolerant control approach for singular systems in the presence of multiplicative failures based on control performance. Reference [19] discusses the design of an AFTCS for spacecraft attitude control with actuator defects, fault estimation errors, and control input limitations.

For sensor errors in air–fuel ratio management of internal combustion engines, fuzzy-logic-based AFTCS was developed in [20] but it consisted only of an active part having computational inefficiency. In [21], AFTCS was proposed for both sensor and actuator faults with mathematical relationship observers and redundant actuators for the anti-surge control system of centrifugal compressors. The approach suggested is less sensitive due to substantial computational cost, and undesirable transients in switching the control strategy from a regular controller to a reconfigurable controller once the problem is effectively diagnosed, which are not prevented throughout this investigation. The applications of FTC can be found in aerial vehicles such as quadcopters and hex copters as well. Fault detection and fault-tolerant control algorithms are suggested in [22] to address the problems of both actuator defects and disturbances in a hex copter. By utilizing sliding mode and a disturbance observer with an FDI, the motor failure is separated to resolve the one or two actuator failures. In [23], a strategy for estimating actuator faults in quadcopters operating in unknown environments is presented. An improved intermediate estimator approach, which can be used to predict actuator failures and system states, was applied to the quadcopter model to avoid the difficulty associated with solving linear matrix inequalities. A sliding mode control based on neural networks is presented in a new study for the attitude and altitude system of a quadcopter amid small perturbations [24]. In [25], the problem of fault-tolerant control for turbofan engines with actuator defects is addressed and solved using Convolution Neural Networks (CNN). During this study, a limited dimensionality is considered; hence, the proposed method is ineffective for the higher dimensionality of turbofan engines. Moreover, the CNN requires a large dataset to produce acceptable results, which makes it slow during the training stage.

The existing FTCS approaches along with their shortcomings are summarized in Table 1.

**Table 1.** FTCS approaches and their drawbacks.

| Technique | Reference | Remarks |
| --- | --- | --- |
| Kalman Filters | [7,8,11] | KFs use a linear state transition and measurement model, hence, less accurate for nonlinear estimation |
| AIO | [9] | Inefficient in terms of computation power |
| Fuzzy Logic | [10,16,20] | The accuracy of these systems is degraded since they frequently use incorrect data and inputs. |
| Linear Regression | [12] | Less accurate estimated values |
| ANN | [15,17] | High-error data points are ignored |
| CNN | [25] | Requires a large dataset |

We can see from the literature that most of the work on fault detection use Kalman filters, lookup tables, linear regression, machine learning, or ANN. For malfunctioning sensors, linear regression gives less accurate sensor data, while the Kalman filters and lookup-table approaches are computationally wasteful. Because gradient descent must be done several times, the ANN-based technique necessitates forward and backward propagation. Because of this, the ANN is sluggish, and the only way to increase its speed is to utilize a GPU, which is a costly option. Hence, we propose three different approaches that can be used to estimate the values for nonlinear sensors. These approaches are PSO,

GA, and NLR, which can be used to design an AFTCS for highly nonlinear sensors of an IC engine for AFR system control.

During this work, the main contribution is the implementation of three estimation techniques for nonlinear sensors, given the constraint that we have a minimum dataset for the prediction of sensor values in case of a fault. The proposed techniques are implemented in such a way that the controller needs much less computational power and computational time to generate the estimated values. Finally, the performance analysis in terms of accuracy and computational power requirements is carried out to prove the superior performance of the proposed techniques. The simulations are carried out by considering an engine speed of 300 r/min, which is the implementation speed of the MATLAB model. The simulation results show the superior performance of NLR over PSO and GA and prove the superior fault tolerance performance, especially for the Manifold Absolute Pressure (MAP) sensor in terms of less oscillatory response, compared to that reported in existing literature.

## 3. Research Methodology

The suggested AFTCS was developed in MATLAB/Simulink using the experimentally proven AFR model [26] of the IC gasoline engine. The results of refining this model to fit the right AFTCS design for a GA, PSO, and NLR-based FDI unit have been reported. The Fault Injection Unit (FIU) sends a fault instruction to each sensor one at a time, while the additional sensors stay normal. The engine speed is set to 300 r/min for this work based on the MATLAB model's design speed, and the FDI unit provides the same value to the controller when the speed sensor fails. The MATLAB model lookup tables [27] are used to extract the data for MAP and throttle sensors at 300 r/min. GA, PSO, and NLR techniques are used to create nonlinear relationships between MAP and throttle using the supplied data. These nonlinear relationships are then used by the FDI unit to estimate the values of the faulty sensors.

### 3.1. Air–Fuel Ratio (AFR) Control

An Internal Combustion (IC) engine is a heat engine that uses air to burn fuel in a combustion chamber. In industrial operations, IC engines are commonly utilized as prime movers. Chemical energy is turned into mechanical rotation, which is then used to power the engines' compressors and alternators. The two types of IC engines are Spark Ignition (SI) and Compression–Ignition (CI). The compression causes combustion in CI engines, whereas spark plugs cause combustion in SI engines. During the combustion phase, the AFR is a measurement of how often air and fuel are mixed in a specified ratio. It is required for better engine performance, reduced fuel consumption, and less pollution. The general architecture of the AFR of an SI IC engine is shown in Figure 1.

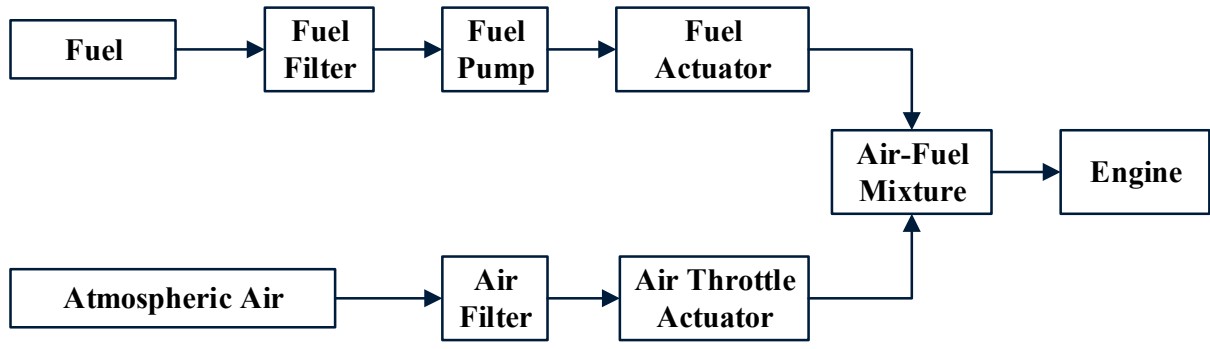

**Figure 1.** The architecture of the air–fuel system of an SI IC Engine [4].

The mathematical equation of AFR is:

$$\text{AFR} = \frac{m_{air}}{m_{fuel}} \tag{1}$$

The chemical equation is given as:

$$25O_2 + 2C_8H_{18} \rightarrow 16CO_2 + 18H_2O + \text{Energy} \tag{2}$$

The AFR is referred to as the stoichiometric ratio in this equation, and its value for gasoline is 14.6:1. During the combustion of fuel, the AFR ranges from 6:1 to 20:1. A rich mixture has a lower value than the stoichiometric ratio, whereas a lean mixture has a higher value than this ratio. The 16.5:1 AFR, for example, is lean, while the 13.7:1 AFR is rich in gasoline. Rich and lean mixtures are also deemed dangerous to the engine because they weaken the catalyst while also lowering engine performance and fuel economy. The value of AFR varies depending on the type of petrol. The value for ethanol amounts to 9:1, methanol to 6.47:1, and hydrogen to 34.3:1. Four sensors are important in preserving AFR control of the SI IC engines:

*Manifold Absolute Pressure (MAP) Sensor:* It gives the controller a precise suction air pressure rating.

*Throttle Sensor:* It provides the controller with an air throttle position.

*Exhaust Gas Oxygen (EGO) Sensor:* It is used to assess the oxygen substance in the drained gas and manages the fuel supply to ensure efficient ignition.

*Speed Sensor:* Measures the rotational speed of the crankshaft of an engine; to provide this speed to the controller, a speed sensor is used.

Because when these sensors fail, the engine shuts down, hence fault tolerance is required. It is desirable to generate virtual redundant sensors in the FDI unit that have a nonlinear response similar to actual sensors. That is why we used the estimation techniques (GA, PSO, and NLR) in AFTCS architecture to approximate the system.

### 3.2. System Modeling

The AFR control can be categorized in a variety of dynamic ranges, such as air dynamics, fuel dynamics, sensor model, and controller design [28]. Here, the formulation for each dynamic is given.

#### 3.2.1. Air Dynamics

Using the ideal air–gas hypothesis and mass conservation theory, the air intake dynamics are characterized as follows:

$$\dot{P}_{in} = \frac{RT_{in}}{v_{in}}\left(\dot{m}_{th} - \dot{m}_{Cyt}\right) + P_{in}\frac{\dot{T}_{in}}{T_{in}} \tag{3}$$

$$\dot{P}_{in} = \Psi(\phi_{th}, P_{in}, T_{in}, N_e) \tag{4}$$

In the above equations, $T_{in}$ is the temperature at the input, $\dot{P}_{in}$ is manifold pressure, and $v_{in}$ is the volume at the input; $\dot{m}_{th}$ and $\dot{m}_{Cyt}$ are used to indicate the mass flow through the valve and into the cylinder, respectively; $R$ represents the gas constant; the engine speed is denoted by $N_e$, and throttle opening position is represented by $\phi_{th}$. The intake temperature's derivative concerning time is meant to be zero.

Now, Equation (4) becomes:

$$\dot{P}_{in} = \dot{k}_{in}\left(\dot{m}_{th} - \dot{m}_{Cyt}\right) \tag{5}$$

$$with \quad \dot{k}_{in} = \frac{RT_{in}}{v_{in}} \tag{6}$$

The volume of air passing through the valve is [29]:

$$\dot{m}_{th} = C_d \frac{P_{id}}{\sqrt{RT_{id}}} S_{es}(\phi_{th})\, g\,(P_r) \tag{7}$$

where $C_d$ is the coefficient of discharge. $P_{id}$ indicates the overhead load pressure. The load ratio $P_r$ then can be defined as $P_r = \frac{P_{in}}{P_{id}}$.

$S_{es}(\phi_{th})$ is the opening area of throttle and can be given as:

$$S_{es}(\phi_{th}) = \sigma_1\{1 - \cos(\sigma_2\phi_{th} + \sigma_3)\} + \sigma_4 \tag{8}$$

Here, $g\ (P_r)$ is a nonlinear quantity and is given as:

$$g\ (P_r) = \begin{cases} \sqrt{\frac{2\gamma}{\gamma-1}}(P_r)^{\frac{1}{\gamma}}\sqrt{\left(1 - P_r^{\frac{\gamma-1}{\gamma}}\right)} \ if \ \ P_r > \left(\frac{2}{\gamma+1}\right)^{\frac{\gamma}{\gamma-1}} \\ \sqrt{\gamma}\left(\frac{2}{\gamma+1}\right)^{\frac{\gamma+1}{2(\gamma-1)}} \qquad if \ \ P_r \leq \left(\frac{2}{\gamma+1}\right)^{\frac{\gamma}{\gamma-1}} \end{cases} \tag{9}$$

Here, $\gamma$ is the air heat ratio and its value is 1.4.

### 3.2.2. Fuel Dynamics

The fuel dynamics can be found in [30] and reported as:

$$\begin{cases} \ddot{m}_{ff}(t) = \frac{1}{\tau_f}(-\dot{m}_{ff}(t) + x\dot{m}_{fi}(t)) \\ \dot{m}_{fv} = (1-x)\dot{m}_{fi}(t) \\ \dot{m}_f(t) = \dot{m}_{fv}(t) + \dot{m}_{ff}(t) \end{cases} \tag{10}$$

where $\tau_f$ represents the fuel vapor process at some fixed amount of time indicated by (s); $\dot{m}_{fi}(t)$ represents the fuel flow injection [kg/s]; the flow of fuel in the cylinders is given by $\dot{m}_f(t)$ [kg/s]; $\dot{m}_{fv}$ the vapor fuel flow [kg/s]; and $\ddot{m}_{ff}(t)$ the liquid mass fuel flow [kg/s]. To produce a more comprehensive model, it is possible to add $\varkappa$ as a vector dependent on the throttle opening or engine r/min $N_e$ [31]. In our case, the second solution is more feasible:

$$\tau_f(N_e) = \sigma_5 N_e^{-\sigma_6} \tag{11}$$

$$\varkappa(N_e) = \sigma_7 + \sigma_8 N_e \tag{12}$$

where $\sigma_5, \sigma_6, \sigma_7, \sigma_8$ are constant parameters. The injector model is given by a linear relationship between the mass fuel flows from the injectors. The AFR now becomes:

$$\lambda_{cyl} = \frac{\dot{m}_{cyl}(t)}{\lambda_s \dot{m}_f(t)} \tag{13}$$

### 3.2.3. Sensor Model

The expression of the lambda ($\lambda$) sensor model can be formed as:

$$\dot{\lambda}(t) = -\frac{1}{\tau_\lambda}\lambda(t) + \frac{1}{\tau_\lambda}\lambda_{cyl}(t - \tau(N_e(t))) \tag{14}$$

Here, $\tau_\lambda$ is the fixed time delay and its value is 0.1 s.

The speed of the engine $N_e(t)$ and time delay $\tau$ are related as follows:

$$\tau(N_e(t)) = \frac{60}{N_e(t)}\left(1 + \frac{1}{n_{cyl}}\right) \tag{15}$$

### 3.2.4. State-Space Representation

To obtain the state-space model, the following equation can be used:

$$\begin{bmatrix} \dot{x}_1 \\ \dot{x}_2 \end{bmatrix} = A\begin{bmatrix} x_1 \\ x_2 \end{bmatrix} + B\begin{bmatrix} u_1 \\ u_2 \end{bmatrix} \tag{16}$$

$$y = C \begin{bmatrix} x_1 \\ x_2 \end{bmatrix} + D \begin{bmatrix} u_1 \\ u_2 \end{bmatrix} \tag{17}$$

$$\begin{cases} \dot{x}_1 = f_1(.)x_1(t) - f_2(.)u(t) \\ \dot{x}_2 = -\frac{1}{\tau_\lambda}\lambda(t) + \frac{1}{\tau_\lambda}\lambda_{cyl}(t - \tau(N_e(t))) \end{cases} \tag{18}$$

with $x_1(t) = \lambda_{cyl}$, $x_2(t) = \lambda(t)$, and $u(t) = \dot{m}_{fi}(t)$:

$$f_1(.) = -\frac{1}{\tau_\lambda(N_e)} - \frac{\dddot{m}_{cyl}}{m_{cyl}(N_e, P_{in})} \tag{19}$$

$$f_2(.) = \lambda_s \frac{\chi(N_e)}{\tau_f(N_e)} m_{cyl}(N_e, P_{in}) \tag{20}$$

confined as follows: $\underline{f_i} \le f_i(\cdot) \le \overline{f_i}$, for $i \in \{1, 2\}$.

3.2.5. Controller Design

To formulate the architecture of the observer, Wang et al. have provided the mathematical model in [32]. We are using the same model to explain the observer's design.

$$\dot{x} = Ax + Bu \tag{21}$$

$$y = Cx + Du \tag{22}$$

$$\dot{\overline{x}} = A\overline{x} + Bu \tag{23}$$

$$\overline{y} = C\overline{x} \tag{24}$$

where "$x$" represents the state vector, "$u$" represents the input vector, and "$y$" represents the output vector. The system matrices are $A$, $B$, $C$, and $D$.

$$(\dot{\overline{x}} - \dot{x}) = A(\overline{x} - x) \tag{25}$$

$$(\overline{y} - y) = C(\overline{x} - x) \tag{26}$$

$$\dot{\overline{x}} = A\overline{x} + Bu + L(\overline{y} - y) \tag{27}$$

In (27), $L$ is the feedback gain.

$$\dot{\overline{x}} - \dot{x} = A(\overline{x} - x) + L(\overline{y} - y) \tag{28}$$

$$(\overline{y} - y) = C(\overline{x} - x) \tag{29}$$

$$\dot{\overline{x}} - \dot{x} = (A + LC)(\overline{x} - x) \tag{30}$$

$$\dot{e}_x = (A + LC)e_x \tag{31}$$

$$(\overline{y} - y) = Ce_x \tag{32}$$

FDI does not declare any fault when "$e_x$" goes to zero. If "$e_x$" is out of bound, an error is identified, and the faulty value will be replaced. The complex structure of AFTCS, and its slow response time due to excessive computations, are two of its main drawbacks [33].

The nonlinear observer design equation we obtain from (27) follows:

$$\dot{\overline{x}}(t) = A\overline{x}(t) + Bu + g(\overline{x}, u, t) + \overline{L}(C\overline{x} - y) \tag{33}$$

where "$g(x, u, t)$" is a nonlinear function and assumed to be globally Lipschitz. $\overline{L}$ is the gain for the nonlinear observer.

Let $e_x(t)$ be the error vector:

$$e_x(t) \triangleq \overline{x}(t) - x(t) \tag{34}$$

From Equations (33) and (34), we obtain:

$$\dot{e}_x = (A + \overline{L}C)e_x(t) + (g(\overline{x}, u, t) - g(x, u, t)) \tag{35}$$

**Lemma 1.** *The error $e_x(t)$ asymptotically approaches 0 if we can obtain a matrix R, X, and scalar $\mu$ such that $R = R^T > 0$ and $\mu > 0$ in order to meet the linear matrix inequality (LMI):*

$$\begin{bmatrix} RA + A^T R + XC + C^T X^T + \mu\lambda^2 I & R \\ R & -\mu I \end{bmatrix} < 0 \tag{36}$$

*The reliability of each sensor is denoted by R.*

The following equation can be used to select the observer gain matrix:

$$\overline{L} = R^{-1}X \tag{37}$$

We can validate the choice of observer gain matrix by evaluating the following Lyapunov function and proving its derivative to be zero:

$$V(t) = e_x^T(t)Re_x(t) \tag{38}$$

Next, we need to check $\dot{V}(x) < 0 \forall x \epsilon D - \{0\}$, as listed below:

$$\begin{aligned} \dot{V}(t) &= e_X^T(RA + R\overline{L}C + A^T R + C^T L^{-T} R)e_x + 2e_x^T R(g(\overline{x}, u, k) - g(x, u, k)) \\ &\leq e_X^T(RA + R\overline{L}C + A^T R + C^T L^{-T} R)e_x + 1/\mu e_x^T R^2 e_x + \mu \|g(\overline{x}, u, k) - g(x, u, k)\|^2 \\ &\leq e_X^T(RA + R\overline{L}C + A^T R + C^T L^{-T} R)e_x + 1/\mu e_x^T R^2 e_x + \mu\lambda^2 \|e_x\|^2 \\ &= e_X^T((RA + R\overline{L}C + A^T R + C^T L^{-T} R) + \mu\lambda^2 I + 1/\mu R^2)e_x \end{aligned} \tag{39}$$

Substituting (38) in (39), we obtain:

$$\dot{V}(t) \leq e_X^T((RA + XC + A^T R + C^T X^T) + \mu\lambda^2 I + 1/\mu R^2)e_x \tag{40}$$

If the inequality is given in (40) exits, $e_x$ asymptotically approaches 0.

$$(RA + XC + A^T R + C^T X^T + \mu\lambda^2 I + 1/\mu R^2) < 0 \tag{41}$$

This inequality becomes equal to (41) by applying the Schur complement, which completes the proof.

**Theorem 1.** *The error $e_x(t)$ approaches to zero exponentially with rate $k/2$ if there exist matrices R, X, and scalars $\mu, k$ such that $R = R^T > 0$ and $\mu, k > 0$ to satisfy the following:*

$$\begin{bmatrix} RA + A^T R + XC + C^T X^T + \mu\lambda^2 I + kR & R \\ R & -\mu I \end{bmatrix} < 0 \tag{42}$$

To prove this, consider (40) and (42), which give

$$\dot{V}(t) \leq -ke_x^T Re_x = -kV(t) \tag{43}$$

Hence, we can write

$$V(t) \leq e_x^T V(0) \tag{44}$$

From (38) we obtain

$$\lambda_{min}(R)\|e_x(t)\|^2 \leq e^{-kt}\lambda_{max}(R)\|e_x(0)\|^2 \tag{45}$$

where $\lambda_{min}$ and $\lambda_{max}$ are the minimum and maximum eigenvalues of $R$, respectively. Hence, we obtain the following norm:

$$\|e_x(t)\| \leq \sqrt{\lambda_{max}(R)/\lambda_{min}(R)} \; e_x(0) \; e^{-kt/2} \tag{46}$$

Coming back to the residual equation:

$$r(t) \hat{=} \|C\overline{x}(t) - y(t)\| \tag{47}$$

$$r(t) \leq \sqrt{\lambda_{max}(R)/\lambda_{min}(R)} \; \|C\|e_x(0) \; e^{-kt/2} \tag{48}$$

$$\|C\|\|e_x\|(0) \approx \|r(0)\| \tag{49}$$

Finally, we can find the following criteria for fault detection of a sensor fault:

$$r(t) \begin{cases} \leq \sqrt{\lambda_{max}(R)/\lambda_{min}(R)} \; \|r(0)\| \; e^{-\frac{kt}{2}}, \; there \; is \; no \; fault \\ > \sqrt{\lambda_{max}(R)/\lambda_{min}(R)} \; \|r(0)\| \; e^{-\frac{kt}{2}}, \; there \; is \; a \; fault \end{cases} \tag{50}$$

Figure 2 depicts the planned AFTCS and its activity. When the system first begins, it tests the sensor values and calculates the sensor-to-observer value threshold $\xi$.

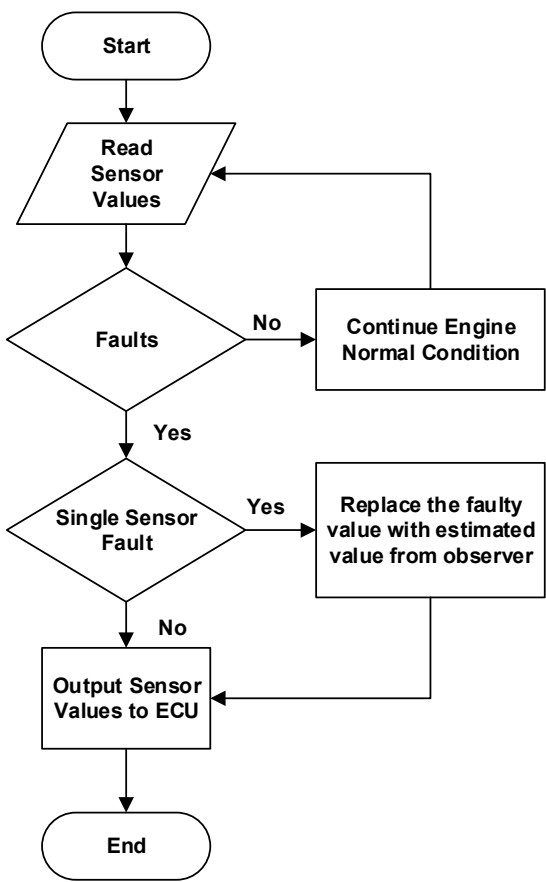

**Figure 2.** Flowchart of proposed AFTCS.

The FDI detects the fault by calculating the residual $e_x$ and comparing it with the threshold ($\xi$) as follows:

If $e_x < \xi$, no fault is present in the sensor.

If $e_x \geq \xi$, a fault is detected in the sensor and will be replaced by the observer output.

If there is no fault, the engine performs as anticipated. The error signal, on the other hand, becomes out of the threshold (10% absolute) if any single sensor fails. The FDI unit feeds the Engine Control Unit (ECU), the approximate value derived from the observer model based on PSO, GA, and NLR to replace the defective sensor value. The model's analytical redundancy is provided by the output of the approximate simulated value of the defective sensor.

This model assumes that the engine runs at a constant speed of 300 r/min. We utilized constant speed in this study since the engines in the process plant run at a constant speed most of the time. Because the article is focused on designing an AFTCS system, load changes and their influence on speed are not examined. The data for the MAP sensor and the throttle sensor at 300 r/min are derived using the available MATLAB model Lookup Tables (LTs). To generate nonlinear relation between the MAP sensor and the throttle sensor, the three approaches are applied. For an estimated value of malfunctioning sensors, the FDI unit uses these nonlinear relationships. It is also supposed that the time spent in switching and reconfiguring is zero seconds. In actuality, the controller computations will be delayed. The research has limitations in that it only examines full sensor failure types, neglecting partial failures, which will be addressed in future research. Many physical problems, such as open circuits owing to wire breakage or weak connections, or burnout due to any type of short-circuiting leading the sensor to produce low output, can cause this issue.

## 4. Estimation Techniques

### 4.1. Genetic Algorithm (GA)

A search heuristic algorithm that was inspired by evolutionary theory is the Genetic Algorithm (GA). GA is frequently used to examine alternatives to optimization problems. For the problems that do not have a proven efficient solution (NP-complete problems) and very large search spaces, we often use genetic algorithm. In GA, the fittest generation has a greater chance of surviving in the following generation since we use GA to build several generations before arriving at the ultimate answer. The current population goes through cycles of crossover and mutation. Each generation has a solution, and the best or most useful generation is ultimately picked as the answer. The GA operations are shown in Figure 3. Based on the values of the population, the GA can be binary or real-coded. In this study, we used the real-coded GA.

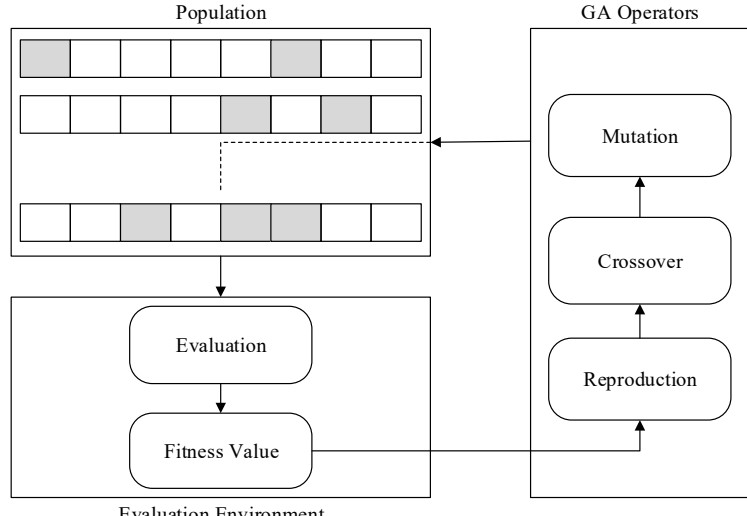

**Figure 3.** Genetic algorithm operations [34].

Uniform crossover is used during this study. The offspring are calculated according to the following equations:

$$y_{1i} = \alpha_i x_{1i} + (1 - \alpha_i) x_{2i} \tag{51}$$

$$y_{1i} = \alpha_i x_{2i} + (1 - \alpha_i) x_{1i} \tag{52}$$

Here, $x_{1i}$ and $x_{2i}$ are parents, $y_{1i}$ and $y_{2i}$ are offsprings, and $\alpha$ is the set of real-valued random numbers which is uniformly distributed over the interval [0, 1]. To obtain better results or expand the search space, $\alpha$ can also be distributed over the interval $[-\gamma, 1 + \gamma]$, where $\gamma$ is a small real number.

To set the mutation parameters, we suppose that the variable $x_j$ is changing to $x_j'$ according to the following relation:

$$x_j' = x_j + \delta \tag{53}$$

$\delta$ is a real-valued random number with any valid continuous probability distribution. In this study, we assumed that $\delta$ has a normal distribution:

$$\delta \sim (\mu, \sigma^2) \tag{54}$$

The variance $\sigma^2$ defines the mutation step size.

Owing to its impressive performance, GA is widely used in optimization and classification problems ranging among fault diagnosis in PV cells [34], OFDM and OQAM synchronization estimation [35], OFDM modulation [36], classification problem solving [37], hyperparameter optimization of power amplifiers [38], optimal resource allocation in cloud computing [39], linear array antenna designing [40], robotic locomotion optimization [41], compressor-based characteristics estimation for a system's slow speed [42], and many more similar problems.

*4.2. Particle Swarm Optimization (PSO)*

PSO is a swarm intelligence and mobility-based stochastic optimization approach. In PSO, the concept of social interaction is used to address issues. It makes use of a swarm of particles (agents) that move around in the search space looking for the best solution. Each particle in the swarm looks for its positional coordinates in the solution space, which are connected to the best solution that the particle has identified so far. $P_{best}$, or personal best, is the term for it. Another best value that the PSO keeps track of is $g_{best}$, or global best. This is the best possible value produced so far by any particle around that particle. In PSO, each particle has a velocity and a position. We need to update the velocities of the particles and move the particles to their new positions. The particles move to their new position according to the following relation:

$$P_i^{t+1} = P_i^t + v_i^{t+1} \tag{55}$$

The velocities are calculated as:

$$V_i^{t+1} = W \cdot V_i^t + c_1 U_1^t (P_{b_1}^t - P_i^t) + c_2 U_2^t (g_b^t - P_i^t) \tag{56}$$

In (56), $W$ is the inertia weight and is a positive constant; this parameter is important for balancing the global search. $V_i$ is the velocity of the particle, $U_1$ and $U_2$ are uniformly distributed random numbers; $c_1$ is the cognitive constant and $c_2$ is a social constant, $P_b$ is the personal best, and $g_b$ is the global best. Clerk and Kennedy [43] defined a general rule for the calculation of inertia weight, cognitive, and social constants in terms of constriction coefficients. The constriction coefficient is given as:

$$\chi = \frac{2k}{\left|2 - \varphi - \sqrt{\varphi^2 - 4\varphi}\right|} \tag{57}$$

Here, kappa ($k$) is constant and $\varphi$ is also a constant whose value is greater than or equal to 4; $\varphi$ can be written as:

$$\varphi = \varphi_1 + \varphi_2 \geq 4 \tag{58}$$

Then, $W$, $c_1$, and $c_2$ can be calculated as:

$$W = \chi$$

$$c_1 = \chi\varphi_1$$

$$c_2 = \chi\varphi_2$$

Similar to GA, PSO is also very popular and used in several optimization problems such as antenna array synthesis, modulation, and demodulation in digital communication, resource allocation in networking, etc.

### 4.3. Nonlinear Regression

Regression is one of the most well-known statistical approaches for predictive data mining. The process of building a model using existing training data that allows you to predict the value of a continuous output variable based on fresh input variable values is known as regression [44]. In statistics, nonlinear regression is a type of regression analysis in which observational data are represented by a nonlinear combination of model parameters that are dependent on one or more independent variables. A system of successive approximations is used to match the results. A statistical model of the nonlinear regression can be given as:

$$y \sim f(X, \beta) \tag{59}$$

This model is used to relate a vector of independent variables, $X$ with the associated dependent variables, vector $y$. In contrast to linear regression, here the function $f$ is nonlinear in the components of the vector $\beta$.

The goal of the model is to obtain the total of the squares to be as small as feasible. The sum of the squares is a metric for measuring how close the $Y$ measurements are to the nonlinear (curved) equation that was used to estimate $Y$. The difference between the fitted nonlinear equation and each $Y$ point in the data set is calculated to obtain it. Following that, each gap must be squared. All the squared figures are brought together at the end. The more the equation matches the data points in the sample, the smaller the sum of these squared figures. Logarithmic functions, trigonometric functions, exponential functions, power functions, Lorenz curves, Gaussian functions, or other similar nonlinear functions are used in nonlinear regression [44]. In the same way that linear regression modeling does, nonlinear regression modeling aims to graphically monitor a certain response from a set of factors. Nonlinear models are more complex to create than linear models since the equation is formed by a series of approximations (iterations) that may come through trial and error [44].

### 5. Results and Discussion

### 5.1. Estimation of Sensor Values

To add the analytical redundancy in the proposed model, the sensor values are replaced by the estimated values in case of a fault. The estimated values are calculated by using PSO, GA, and NLR.

5.1.1. MAP Estimation

The use of PSO, GA, and NLR requires a fitness function. To formulate the fitness function for the MAP estimation, the sum of the least-squares method is used. An exponential equation given by (60) is used to predict the values of MAP sensors in the case of a fault:

$$y_e = 1 - e^{-ax^b} + ce^{-x} + \frac{d}{x^2 - 1} \tag{60}$$

Here, $y_e$ is the estimated MAP value; $x$ is throttle angle; and $a$, $b$, $c$, and $d$ are unknown constants. The difference between the estimated MAP value ($y_e$) and the actual MAP value ($y$) is calculated and the sum of the difference over the entire range of $x$ is computed to find the fitness function $f_M$. The expression for the fitness function is given as:

$$f_M = \frac{1}{2n} \sum_{i=1}^{n} (y_i - y_{ei})^2 \tag{61}$$

Here, $n$ = 17 and the values of $x$ and $y$ are given in Table 2.

**Table 2.** Throttle angle ($x$) and MAP values ($y$) at 300 r/min [27].

| $x$ (Throttle Angle) | 0° | 3° | 6° | 9° | 12° | 15° | 18° | 21° | 24° |
|---|---|---|---|---|---|---|---|---|---|
| $y$ (MAP Value) | 0.091 | 0.114 | 0.191 | 0.329 | 0.545 | 0.745 | 0.857 | 0.915 | 0.946 |
| $x$ (Throttle Angle) | 27° | 30° | 35° | 46° | 57° | 68° | 79° | 90° | |
| $y$ (MAP Value) | 0.964 | 0.975 | 0.985 | 0.994 | 0.997 | 0.998 | 0.999 | 0.999 | |

The fitness function given by (61) is then minimized by using PSO, GA, and NLR to find the best possible values of the unknown constants $a$, $b$, $c$, and $d$. All three optimization techniques were implemented in MATLAB and executed on a Core i7 processor with a RAM of 8 GB. PSO was executed with a swarm size of 1500, maximum iterations were kept at 200, and the values of $\varphi_1$ and $\varphi_2$ were set at 2.1 each; the values of kappa ($k$) and damping ratio of inertia coefficient were assumed to be 1.1 and 0.9 respectively. While implementing the GA, the simulated binary crossover is used instead of blend crossover with parameters $\beta$ = 2.1 and distribution index $\eta_C$ = 2.1. The other parameters $\gamma$, $\mu$, and $\sigma$ were set at 0.1, 0.015, and 1, respectively. The population size was set at 500 with 150 iterations. To implement the NLR, the residual sum of squares (RSS) is computed and then the unknown constants were found for which the RSS is minimum. With the abovementioned settings, all three estimations techniques returned the unknown constants and the estimated MAP values, which are given in Tables 3 and 4, respectively.

**Table 3.** Values of unknown constants found by PSO, GA, and NLR for MAP estimation.

| Constant | PSO | GA | NLR |
|---|---|---|---|
| a | 0.0048 | 0.0106 | 0.0042 |
| b | 2.0909 | 1.7589 | 2.1109 |
| c | 0.4797 | 0.2606 | 0.4897 |
| d | 0.4084 | 0.1690 | 0.3883 |

To examine the performance of the individual algorithms, the line fit plots were made, and then the statistics such as residual sum of squares (RSS), mean square error (MSE), $R^2$, adjusted $R^2$, standard error of estimation (SEE), and optimization times were recorded and given in Table 5. The line fit plots are shown in Figure 4.

**Table 4.** Estimated MAP values with PSO, GA, and NLR.

| Throttle Angle | Actual MAP Value | Estimated MAP Values | | |
|---|---|---|---|---|
| | | PSO | GA | NLR |
| 0° | 0.091 | 0.0813 | 0.0917 | 0.0913 |
| 3° | 0.114 | 0.1216 | 0.1046 | 0.1138 |
| 6° | 0.191 | 0.1954 | 0.2246 | 0.1792 |
| 9° | 0.329 | 0.3805 | 0.3985 | 0.3542 |
| 12° | 0.545 | 0.5791 | 0.5683 | 0.5482 |
| 15° | 0.745 | 0.7475 | 0.7113 | 0.7190 |
| 18° | 0.857 | 0.8666 | 0.8194 | 0.8449 |
| 21° | 0.915 | 0.9381 | 0.8940 | 0.9243 |
| 24° | 0.946 | 0.9749 | 0.9415 | 0.9675 |
| 27° | 0.964 | 0.9913 | 0.9696 | 0.9879 |
| 30° | 0.975 | 0.9975 | 0.9851 | 0.9962 |
| 35° | 0.985 | 1.0000 | 0.9961 | 0.9998 |
| 46° | 0.994 | 1.0002 | 0.9999 | 1.0002 |
| 57° | 0.997 | 1.0001 | 1.0000 | 1.0001 |
| 68° | 0.998 | 1.0001 | 1.0000 | 1.0001 |
| 79° | 0.999 | 1.0001 | 1.0000 | 1.0001 |
| 90° | 0.999 | 1.0001 | 1.0000 | 1.0000 |

**Table 5.** Estimation statistics for MAP estimation.

| Parameter | PSO | GA | NLR |
|---|---|---|---|
| Residual Sum of Squares (RSS) | 0.007 | 0.010 | 0.003 |
| Mean Square Error (MSE) | 0.0004 | 0.0006 | 0.0002 |
| Coefficient of Determination ($R^2$) | 0.99 | 0.99 | 0.99 |
| Adjusted $R^2$ | 0.96 | 0.92 | 0.98 |
| Standard Error of Estimation (SEE) | 0.02 | 0.024 | 0.01 |
| Optimization Time (t) in seconds | 6.71 | 4.25 | 0.46 |

It is evident from Figure 4 that the estimated values generated by NLR are very close to the actual values of the MAP sensor. It is further supported by entries in Table 4. From Table 5, it can be noted that RSS, MSE, $R^2$, and adjusted $R^2$ values for NLR are much better than PSO and GA. It can be seen in Table 5 that the $R^2$ values for all three techniques are the same (0.99), but the adjusted $R^2$ of NLR is very impressive (0.98) as compared to GA (0.92) and PSO (0.96). PSO and NLR produce nearly the same results. Moreover, the SEE value for NLR is 0.01 as compared to 0.02 and 0.024 in PSO and GA, respectively. It proves the higher accuracy of NLR over PSO and GA. To further justify the claim of the effectiveness of NLR, the optimization times are also calculated by using MATLAB, and it can be noted from Table 5 that the optimization time of NLR is shorter (0.46 s) when compared to PSO (6.71 s) and GA (4.25 s), which makes the NLR useful for time-sensitive applications and controllers with less computation power.

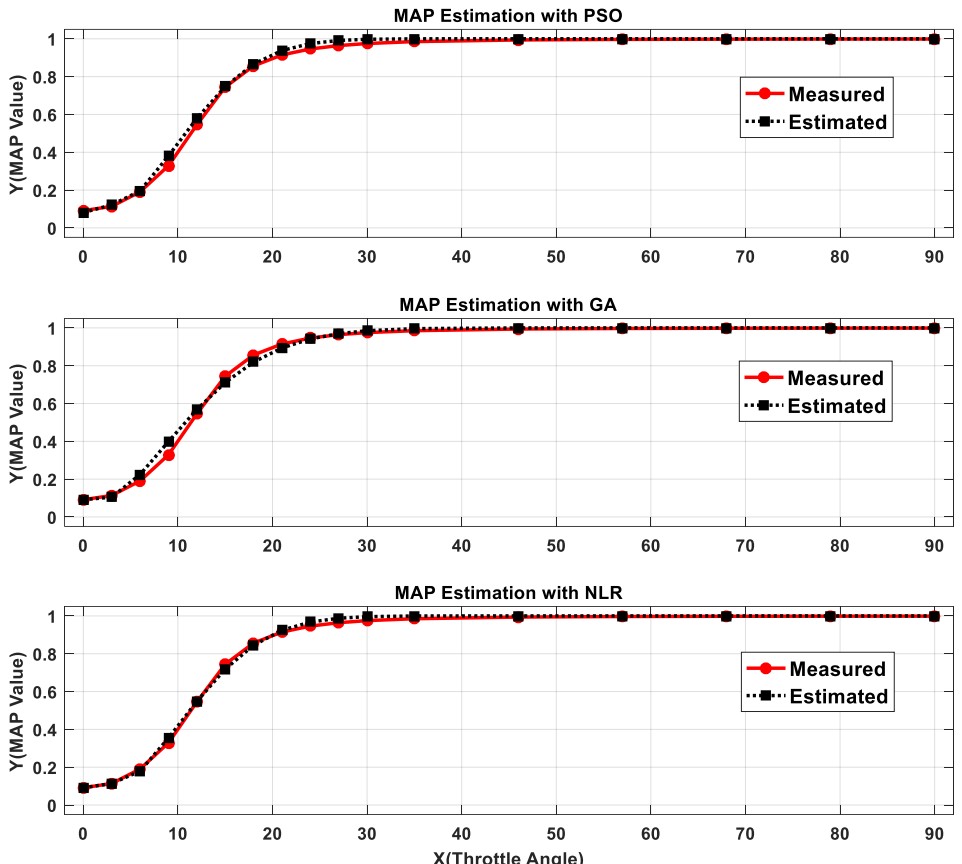

**Figure 4.** Estimated MAP values with PSO, GA, and NLR.

5.1.2. Throttle Estimation

The use of PSO, GA, and NLR requires a fitness function. To formulate the fitness function for the throttle estimation, the sum of the least-squares method is used. A 6$^{th}$-degree polynomial given by (62) is used to predict the values of throttle sensors in the case of a fault:

$$y_e = a_6 x^6 + a_5 x^5 + a_4 x^4 + a_3 x^3 + a_2 x^2 + a_1 x + a_0 \qquad (62)$$

Here, $y_e$ is the estimated throttle value; $x$ is MAP value; and $a_6, a_5, a_4, a_3, a_2, a_1$, and $a_0$ are unknown constants. The difference between estimated throttle value ($y_e$) and actual throttle value ($y$) is calculated and the sum of the difference over the entire range of $x$ is computed to find the fitness function $f_M$. The expression for the fitness function is given as:

$$f_M = \frac{1}{2n} \sum_{i=1}^{n} (y_i - y_{ei})^2 \qquad (63)$$

For throttle estimation, $n = 19$, and the values of $x$ and $y$ are given in Table 6.

The fitness function given by (63) is then minimized by using PSO, GA, and NLR to find the best possible values of the unknown constants $a_0$ through $a_6$. All three optimization techniques were implemented in MATLAB and executed on a Core i7 processor with a RAM of 8 GB. PSO was executed with a swarm size of 1800, maximum iterations were kept at 500, and the values of $\varphi_1$ and $\varphi_2$ were set at 2.9 and 1.5, respectively; the values of kappa ($k$) and damping ratio of inertia coefficient were assumed to be 1.5 and 0.2. While implementing the GA, the simulated binary crossover is used instead of blend crossover with parameters $\beta = 3.8$ and distribution index $\eta_C = 2.1$. The other parameters $\gamma$, $\mu$, and $\sigma$ were set at 0.25, 0.02, and 0.5, respectively. The population size was set at 400 with 100 iterations. To implement the NLR, the residual sum of squares (RSS) is computed and then the unknown constants were found for which the RSS is minimum. With the

abovementioned settings, all three estimation techniques returned the unknown constants and the estimated throttle values, which are given in Tables 7 and 8, respectively.

**Table 6.** MAP value ($x$) and throttle angles ($y$) at 300 r/min [27].

| $x$ (MAP Value) | 0.05 | 0.10 | 0.15 | 0.20 | 0.25 | 0.30 | 0.35 |
|---|---|---|---|---|---|---|---|
| $y$ (Throttle Angle) | 0° | 1.98° | 4.69° | 6.26° | 7.47° | 8.48° | 9.36° |
| $x$ (MAP Value) | 0.40 | 0.45 | 0.50 | 0.55 | 0.60 | 0.65 | 0.70 |
| $y$ (Throttle Angle) | 10.13° | 10.82° | 11.45° | 12.06° | 12.70° | 13.40° | 14.19° |
| $x$ (MAP Value) | 0.75 | 0.80 | 0.85 | 0.90 | 0.95 | | |
| $y$ (Throttle Angle) | 15.11° | 16.24° | 17.75° | 20.03° | 24.50° | | |

**Table 7.** Values of unknown constants found by PSO, GA, and NLR for throttle estimation.

| Constant | PSO | GA | NLR |
|---|---|---|---|
| $a_6$ | 52.92 | 45.88 | 806.64 |
| $a_5$ | −57.10 | −28.79 | −2144.63 |
| $a_4$ | 59.85 | 32.90 | 2157.77 |
| $a_3$ | −60.03 | −56.51 | −956.47 |
| $a_2$ | 0.001 | 2.40 | 129.75 |
| $a_1$ | 35.11 | 35.56 | 40.77 |
| $a_0$ | −1.03 | −1.26 | −2.34 |

**Table 8.** Estimated throttle values with PSO, GA, and NLR.

| MAP Value | Actual Throttle Value | Estimated Throttle Values | | |
|---|---|---|---|---|
| | | PSO | GA | NLR |
| 0.05 | 0° | 0.72° | 0.52° | −0.08° |
| 0.10 | 1.98° | 2.43° | 2.27° | 2.28° |
| 0.15 | 4.69° | 4.06° | 3.95° | 4.41° |
| 0.20 | 6.26° | 5.59° | 5.54° | 6.17° |
| 0.25 | 7.47° | 7.00° | 7.01° | 7.55° |
| 0.30 | 8.48° | 8.27° | 8.33° | 8.60° |
| 0.35 | 9.36° | 9.38° | 9.48° | 9.42° |
| 0.40 | 10.13° | 10.34° | 10.47° | 10.10° |
| 0.45 | 10.82° | 11.14° | 11.28° | 10.73° |
| 0.50 | 11.45° | 11.80° | 11.93° | 11.37° |
| 0.55 | 12.06° | 12.36° | 12.46° | 12.05° |
| 0.60 | 12.70° | 12.85° | 12.90° | 12.76° |
| 0.65 | 13.40° | 13.35° | 13.35° | 13.49° |
| 0.70 | 14.19° | 13.95° | 13.89° | 14.25° |
| 0.75 | 15.11° | 14.78° | 14.67° | 15.08° |
| 0.80 | 16.24° | 16.00° | 15.87° | 16.13° |
| 0.85 | 17.75° | 17.81° | 17.71° | 17.68° |
| 0.90 | 20.03° | 20.48° | 20.47° | 20.20° |
| 0.95 | 24.50° | 24.32° | 24.50° | 24.44° |

To examine the performance of the individual algorithms, the line fit plots were made, and then the statistics such as residual sum of squares (RSS), mean square error (MSE), $R^2$, adjusted $R^2$, standard error of estimation (SEE), and optimization times were recorded and given in Table 9. The line fit plots are shown in Figure 5.

**Table 9.** Estimation statistics for throttle estimation.

| Parameter | PSO | GA | NLR |
|---|---|---|---|
| Residual Sum of Squares (RSS) | 2.65 | 3.02 | 0.28 |
| Mean Square Error (MSE) | 0.14 | 0.16 | 0.01 |
| Coefficient of Determination ($R^2$) | 0.996 | 0.995 | 0.999 |
| Adjusted $R^2$ | 0.94 | 0.93 | 0.99 |
| Standard Error of Estimation (SEE) | 0.40 | 0.42 | 0.10 |
| Optimization Time (t) in seconds | 26.63 | 3.03 | 0.097 |

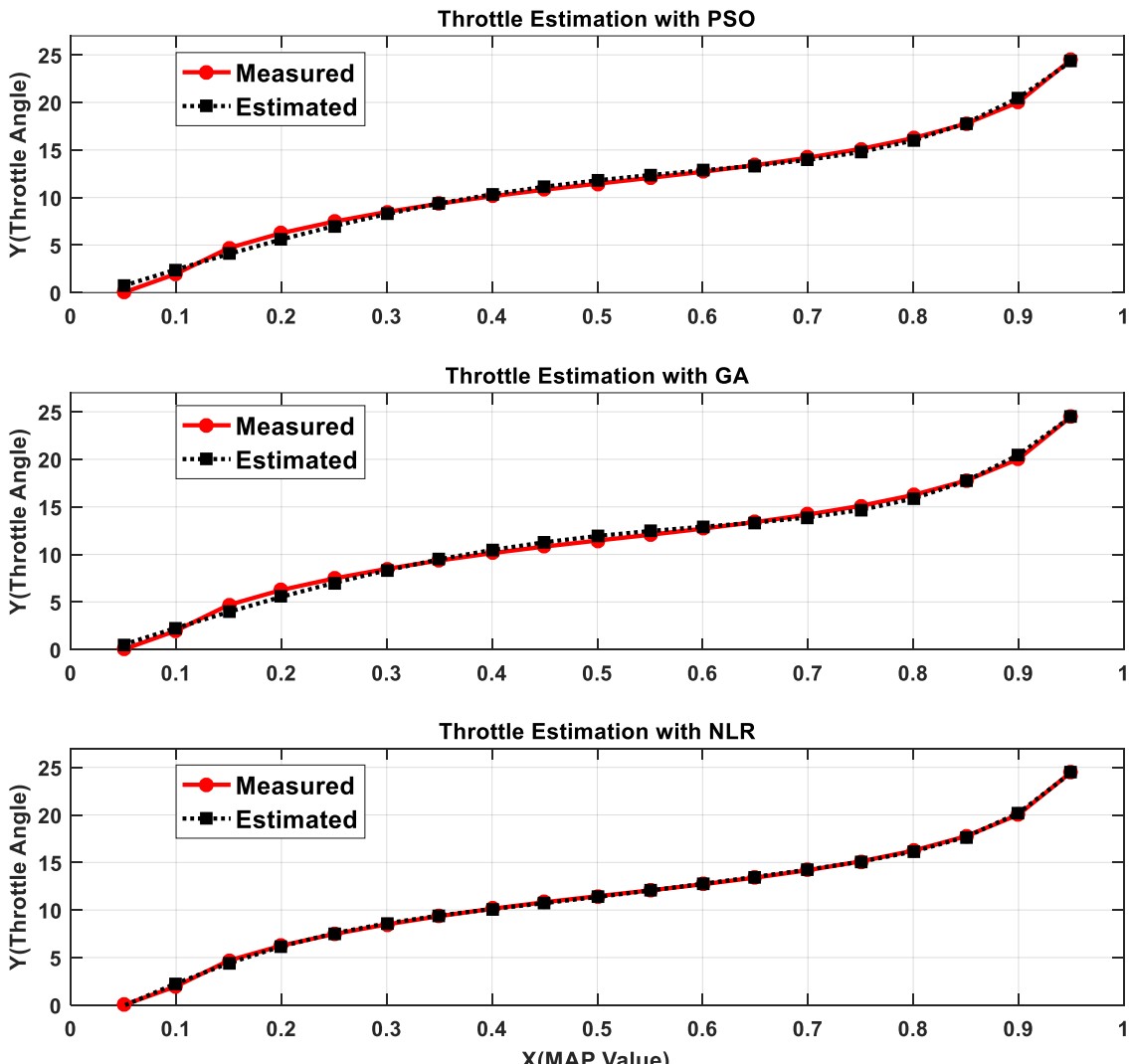

**Figure 5.** Estimated throttle values with PSO, GA, and NLR.

It is evident from Figure 5 that the estimated values generated by NLR are very close to the actual values of the throttle sensor. It is further supported by entries in Table 8. From Table 9, it can be noted that RSS, MSE, $R^2$, and adjusted $R^2$ values for NLR are much

better than those for PSO and GA. It can be seen in Table 9 that the $R^2$ values for all three techniques are the same (0.99), but the adjusted $R^2$ of NLR is very impressive (0.99) as compared to GA (0.93) and PSO (0.94). PSO and NLR produce nearly the same results. Moreover, the SEE value for NLR is only 0.1 as compared to 0.4 and 0.42 in PSO and GA, respectively. It proves the higher accuracy of NLR over PSO and GA. To further justify the claim of the effectiveness of NLR, the optimization times are also calculated by using MATLAB, and it can be noted from Table 9 that the optimization time of NLR is shorter (0.097 s) as compared to PSO (26.63 s) and GA (3.03 s), which makes the NLR useful for time-sensitive applications and controllers with less computation power. Both PSO and GA produce identical results, but it is worth noticing that PSO consumes much more time as compared to GA. This fact makes GA the second-best option for this optimization, after NLR.

### 5.2. MATLAB Simulation Results

The proposed AFTCS for the AFR controller was implemented in a MATLAB IC gasoline engine model. The FDI block is used to identify, separate, and substitute inaccurate parameters with expected values. As shown in Section 5.1, the NLR provides more accurate results by consuming less computation power; thus, the observer model in the FDI unit was built with the nonlinear regression relationships. The AFTCS phase is simulated with failures, based on input from the other functional sensors in the FDI unit; the FLC observer develops a new estimated value, which is delivered to the controller. The defect is injected into each sensor, one at a time by the FIU, which keeps the others healthy. The mixture AFR ratio is kept at 14.6 in normal circumstances, while it decreases to 11.7 under faulty situations (rich mixture). Despite the degradation, the system maintains stability, which is compatible with AFTCS design philosophy [45]. Figure 6 shows how the proposed AFTCS performs in both normal and abnormal scenarios.

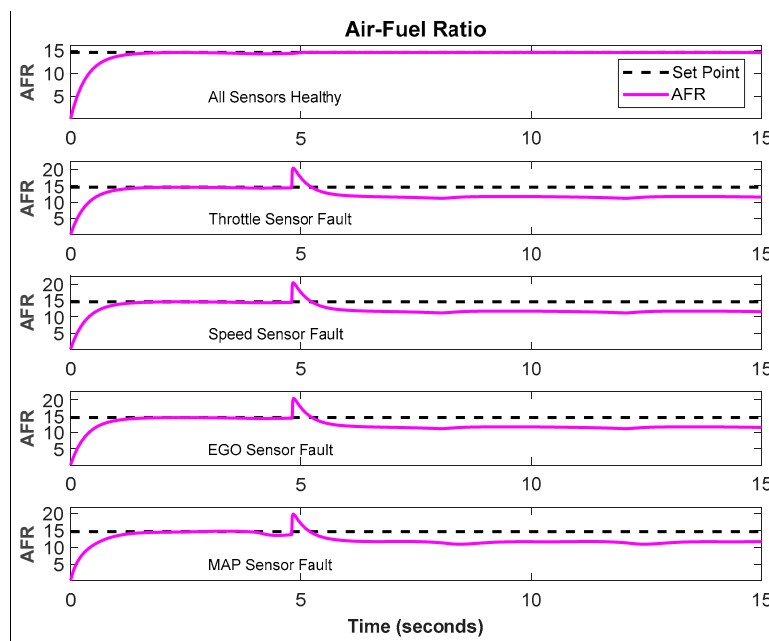

**Figure 6.** AFTCS performance for AFR control of the IC engine.

The AFR deteriorated to 11.7 in the steady state with a fault in any one sensor with the AFTCS portion alone, as shown in Figure 6. It can be noted from Figure 6 that the engine response is less oscillatory as compared to results reported in [12], especially when the fault is introduced in the MAP sensor. This is because the estimated values generated by NLR are very close to the actual sensor values, which validates the analytical redundancy and proves the effectiveness of NLR.

### 5.3. Reliability Analysis

In this section, the probabilistic reliability analysis was carried out for our proposed AFTCS model. Let $R$ represents the reliability of each sensor, then the probability of failure of these components follows:

$$P_{CF} = 1 - R \tag{64}$$

where $P_{CF}$ represents the probability of component failure.

Without fault tolerance, the engine shuts down due to a fault in any of the four sensors.

Let $P_{SF}$ represent the probability of system failure and $R_{System}$ represents the reliability of the overall system, then we have

$$P_{SF} = 4(1 - R) \tag{65}$$

$$R_{System} = 1 - P_{SF} \tag{66}$$

$$R_{System} = 1 - 4\,(1 - R) \tag{67}$$

If $R = 0.9$, then $R_{System} = 1 - 4\,(1 - 0.9) = 0.6$

$$R_{System} = 60\%$$

The proposed AFTCS prevents the engine from shutting down in the event of a malfunction in a single sensor. However, if two sensors fail at the same moment, the engine will shut down. Hence,

$$P_{SF} = {}^4_2C(1 - R)(1 - R) \tag{68}$$

$$P_{SF} = 6\,(1 - R)^2 \tag{69}$$

$$R_{System} = 1 - P_{SF} \tag{70}$$

$$R_{System} = 1 - 6\,(1 - R)^2 \tag{71}$$

If $R = 0.9$, then $R_{System} = 1 - 6\,(1 - 0.9)^2 = 0.94$

$$R_{System} = 94\%$$

Thus, the reliability of the system improves by 34% with the proposed AFTCS for sensor faults.

From the abovementioned MAP estimation results, throttle estimations results, MATLAB simulation results, and reliability analysis, we can say that the proposed methods, and especially the nonlinear regression, are more efficient as compared to other techniques because:

- The estimated values generated by PSO, GA, and NLR are very close to the actual MAP and throttle values. The standard error of estimation is less than 0.02 for MAP estimation and less than 0.4 for throttle estimation.
- The computational power requirements are much less for these techniques.
- The computational time needed by these techniques is much less, which makes these methods efficient in terms of response time in the case of a sensor fault.

## 6. Conclusions

In this article, a comparative study was reported for the design of an advanced AFTCS for AFR control of an IC engine to achieve greater reliability and prevent costly shutdowns due to sensor faults. These comparisons consisted of three estimations techniques: PSO, GA, and NLR. All three techniques were used to calculate the estimated values for MAP and throttle values of an IC engine in the case of a single sensor failure. The accuracy and the computation speed of these algorithms were measured by using MATLAB and it was proved that NLR produces more accurate results very quickly as compared to PSO and GA. An AFTCS was also proposed for the IC engine. In the proposed AFTCS, the FDI unit was

built with a nonlinear regression-based observer unit. The model was implemented in the MATLAB/Simulink environment and tested for various sensor faults. The results prove superior fault tolerance performance for sensor faults of the AFR control system, especially for the MAP sensor in terms of less oscillatory response as compared to those reported in existing literature. It was also proved that the proposed model is suitable for time-critical applications and controllers with limited computation power.

Since we have considered full-type faults of the sensors as a limiting case of study, future experimental work may include the AFTCS design for partial-type faults of the sensors with experimentation, such as hardware-in-the-loop testing.

**Author Contributions:** Conceptualization, A.A.A.; Formal analysis, M.S.I. and M.B.Q.; Funding acquisition, T.A.; Investigation, M.J.; Methodology, M.S.I. and A.A.A.; Project administration, A.A.A. and M.J.; Resources, M.B.Q.; Software, M.B.Q.; Supervision, T.A.; Validation, S.A.; Visualization, S.A.; Writing—original draft, M.S.I.; Writing—review & editing, A.A.A., S.A. and M.J. All authors have read and agreed to the published version of the manuscript.

**Funding:** This research was supported by the Deanship of Scientific Research at Najran University for funding this work under the Research Groups Funding program, grant code (NU/RG/SERC/11/5).

**Data Availability Statement:** All necessary data used in the research are available in the open-access Mathworks model [26].

**Acknowledgments:** The authors are thankful to the Deanship of Scientific Research at Najran University for funding this work under the Research Groups Funding program, grant code (NU/RG/SERC/11/5).

**Conflicts of Interest:** The authors declare no conflict of interest.

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
