# Peer review of "A Comparative Study of Design of Active Fault-Tolerant Control System for Air–Fuel Ratio Control of Internal Combustion Engine Using Particle Swarm Optimization, Genetic Algorithm, and Nonlinear Regression-Based Observer Model"

_applsci, doi:10.3390/app12157841_

Round 1

Reviewer 1 Report

applsci-1780718-peer-review-v1

Title:  A Comparative Study of Design of Active Fault-Tolerant Control System
for Air-fuel Ratio Control of Internal Combustion Engine Using Particle Swarm
Optimization, Genetic Algorithm, and Nonlinear Regression-Based Observer Model

.

In this paper, author shares a study on the comparative study on particle swarm optimization, genetic algorithm, and nonlinear regression-based observer model. Why it is proposed; it is not clear. This is an ordinary manuscript.  There is no strong result and application in this manuscript.  Further, comments are the following:

1. The abstract isn't sufficiently concise and informative.

2. The purpose of the article doesn't clearly state in the introduction.

3. The article achieve doesn't declare the purpose. Many references are not validated.

4. The article doesn't show clarity of presentation.

5. The English and syntax of the article are not satisfactory.

6. The document is not concise.

***

Reviewer 2 Report

Please see my comments as the file below

Reviewer 3 Report

The manuscript presents a comparative study of design of Active Fault-Tolerant Control System for air-fuel ratio control of internal combustion engine. The following comments are given to further improve the manuscript quality:

1.    Avoid lumping references, e.g. 1-3 and similar. Instead summarize the main contribution of each referenced paper in a separate sentence and/or cite the most recent and/or relevant one.

2.    The authors should more clearly explain why they have chosen these three AFTCS strategies for their study and why they turn out to be better then others? 

In overall the contribution of the manuscript is acceptable but it needs a minor revision.

Round 2

Reviewer 1 Report

applsci-1780718-peer-review-v2

Title:  A Comparative Study of Design of Active Fault-Tolerant Control System
for Air-fuel Ratio Control of Internal Combustion Engine Using Particle Swarm
Optimization, Genetic Algorithm, and Nonlinear Regression-Based Observer Model

.

The revised paper has not improved. All comments are not solved. I do not see anything new such as the appropriateness, novelty and general significance. There are also several technical content and quality issues. I regret to recommend to reject this paper.

However, the following comments may be useful to the authors: 

1.        In Section 1, Author claims <Fault tolerance refers to a system's capacity to continue operating under malfunctioning conditions>. No citation was found to your arguments and problem and motives. Please support them with citations.

2.        Novelty not clear. A comprehensive table for literature survey should be presented by the authors to show the literature review based on their assumptions, methods, and results.

3.        Even though the authors claim a native English speaker has proofread the manuscript, there are still too many grammatical issues in the article. Many expressions and languages used are inappropriate.

4.        The manuscript is poorly structured. Why is a literature review jam-packed within the introduction and not presented as a standalone section?

5.        The article doesn't show clarity of presentation.

***

Reviewer 2 Report

Authors handled all my comments. This paper is acceptable.
